

# Mapping Transboundary Climate Change Risk: the case study of the Trinational Metropolitan Area Upper Rhine Area

Nils Riach[1], Nicolas Scholze[1], and Rüdiger Glaser[1]

[1]1Physical Geography, Institute of Environmental Social Sciences and Geography, University of Freiburg, Germany

**Correspondence:** Nils Riach (nils.riach@geographie.uni-freiburg.de)

**Abstract.** In this study we examine the spatial patterns of risk towards climate change and climatic extremes in the "Trinational Metropolitan Region Upper Rhine" (TMU). Following the concept of risk analysis, we identify the regional dimension of climatic stressors in relation to the socio-economic dimension. We present an indicator-based assessment, which operationalizes risk as a product of its subcomponents climatic stressors, spatial occurrence, sensitivity and impact. We map them individually and aggregate them into a composite index. We also address the specific challenges of the trinational situation, which has an impact on the availability, homogeneity and resolution of comparable data sets. We show that risk can be approximated and mapped despite the uncertainties and additionally we explore to what extent the subcomponents contribute to the overall index. The results show differentiated spatial patterns of risk with cross-border clusters i.e. transnational corridors. Risk is amplified depending on the driving climate scenario for 2021-2050, 2041-2070 and 2071-2100, and increases during the course of the century, especially in the transnational metropolitan corridors of the TMU. Further focus on transnational spatial planning and cooperation is needed in future adaption research and practice.

## 1 Introduction

In vulnerability and risk research, case studies are helpful in gaining an understanding of the determinants of vulnerability, their interactions and how adaptive capacity can be enhanced (Ford et al., 2010). Consequently, the IPCC's (2021) latest report (WGI AR6) also calls for a more in-depth investigation of climate-related risks and their implications for adaptation planning on the regional and local scale. On this scale, vulnerability assessments are most commonly carried out within specific administrative units, e.g. national or subnational borders. There are several good reasons for this, the most prominent being data consistency and homogeneity. However, the transboundary nature of vulnerability has often received insufficient attention (Birkmann et al., 2021b).This makes it difficult to compare areas with similar (climatic) characteristics but different national affiliations. As an example, the climatic conditions in the transnational (German-French-Swiss) Upper Rhine region are comparable (Parlow and Gossmann, 2001), but the assessment of vulnerabilities differ greatly, depending on the national context. While the region is considered a hotspot for the German context (Kahlenborn et al., 2021), it only occupies a middling position from the French perspective (SDES, 2020). In addition, coping strategies towards climatic events can be quite different, as (Himmelsbach et al., 2015) show for flood risks in the region. Although the cross-border perspective can certainly provide new insights, additional



methodological difficulties arise from comparative analyses. They are mostly linked to different origins of data sets, thus reducing availability and limiting comparability.

In this case study we present an empirical GIS-based assessment climate related socio-economic risks on the community scale of the Trinational Metropolitan Area Upper Rhine (TMU). We introduce a consistent methodology by applying standard criteria to potential indicators in order to ensure data quality. In doing so, we are able to explore the transnationality of risk and

its' subcomponents despite the challenging data situation.

## 1.1 Climate change in the TMU

The Trinational Metropolitan Area Upper Rhine (fig. 1) in the centre of Europe comprises the two French départements Bas-Rhin and Haut-Rhin; the German regions of Baden and southern Palatinate and the Swiss cantons of Basel-Stadt, Basel-Landschaft, Jura, Solothurn and Aargau with an approximated population of 6.18 Mio (2016) (Deutsch-Französisch-Schweizerische

Oberrheinkonferenz, 2018). Different landscapes from the Upper Rhine lowlands (approximately 100 m a.s.l) to the medium-high mountain ranges (Feldberg: 1493 m a.s.l.) contribute to a complex orographic and ecologic situation that manifests itself in different climates. The main climatic stressors in the future include a significant increase in heat stress, intensification of heavy rain events and winter precipitation with a correspondingly increased flood risk, increase of drought frequency and severity as well as the decrease of winter snow cover (Parlow et al., 2006; Ouzeau et al., 2014; Riach et al., 2019).The different

landscapes show a significant change of the impact of the main climatic stressor. While the lowland is impacted by heat stress and increasing temperatures, the mountainous areas are more affected by wind, snow and coldness.

The TMU is further characterized by strong cross-border and transnational economic linkages but differing political, administrative, cultural and legal conditions. The polycentric metropolitan area contains highly urbanized agglomerations and corridors around the core cities of Basel, Strasbourg, Karlsruhe, Freiburg and Mulhouse, which are linked by a dense network

of small and medium-sized towns along the main traffic axes. While the densely populated areas are a characteristic for the lowlands, the peripheral and mountainous areas are sparsely populated. The economic structure includes both internationally active companies and numerous medium-sized enterprises, some of which are referred to as hidden champions (Scholze et al., 2020).

Climate change poses a multitude of challenges to all the economic sectors (Zebisch et al., 2005; Stock et al., 2009; Nies,

M., Apfel, D., 2011; Min, 2015; Brasseur et al., 2017; Kahlenborn et al., 2021). However, few systematic assessments of business vulnerability exist (Lo et al. 2021). Generally, small and medium-sized enterprises (SME) are considered vulnerable to climate change, but this is highly sector-specific (Scholze et al. 2018; Averbeck et al. 2019). Adaptation measures for the different economic sectors are not always transferable and general statements are difficult to make. Assessments, which do not examine individual sectors but rather the entire economic spectrum, are therefore based on quite general indicators (Scholze

et al., 2020).





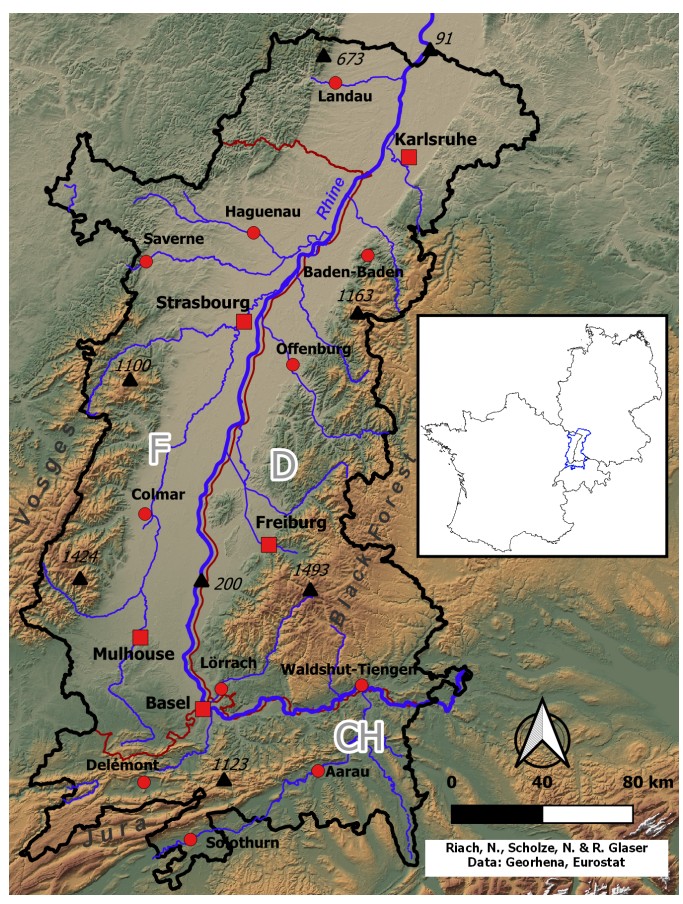

**Figure 1.** The Trinational Metropolitan Area Upper Rhine (TMU) in its Central European context (Scholze et al. 2020).

## 1.2 Climate vulnerability and risk

A major issue in climate change vulnerability and risk literature is the inconsistent use of terminology, which is why a more in-depth explanation is in order. The reason for this plurality is that vulnerability and risk concepts have been developed and applied in numerous academic disciplines. Widely acknowledged interpretations of vulnerability have been put forward, for ex-

ample by Watts and Bohle (1993); Blaikie et al. (1994); Cutter (1994); Wisner et al. (2004); Birkmann (2006); Füssel and Klein (2006); Füssel (2007); IPCC (2007, 2012). Birkmann (2013b) points out that, despite being contested and differently framed, the main achievement of the concept of vulnerability lies within its ability to include social factors and societal structures in the construction of risk and of adaption options. In climate research, the concept of vulnerability intends to inform policies aiming towards the reduction of risk associated with climate change (Birkmann, 2013a). Literature reviews have clustered

the extensive applications of conceptual frameworks differently over the years (Birkmann, 2013b; Jurgilevich et al., 2017; de Sherbinin et al., 2019), but all agree on a paradigm shift between the IPCC's fourth assessment report (AR4, 2007) and the so called SREX (IPCC 2012). In the former, vulnerability is the function of exposure to climatic stimuli, sensitivity and



adaptive capacity of a system. The latter, incorporates sensitivity and adaptive capacity as vulnerability, thus rendering as a subcomponent of a disaster risk perspective. Here, vulnerability, exposure and climatological/hazardous events are more clearly

separated. Exposure in the AR4 framework is a subcomponent of vulnerability. The SREX framework's risk perspective, which has also been adopted by the AR5, distinguishes between exposure and vulnerability and thus represents social vulnerability (de Sherbinin et al., 2019). While the AR4 regards vulnerability as a pre-existing condition, risk in the SREX/AR5 is seen as an outcome. The former being used to explore, understand and trace the causes of vulnerability and often serving for the identification of 'hot-spots' or most vulnerable population groups and areas, the latter, in contrast, characterizes the net result after

adaption measures have been taken, thus, methodological, advancing the previous approach by simulating adaption measures (Jurgilevich et al., 2017).

No single conceptual approach can capture and explain vulnerability and risk comprehensively and the context dependency makes generalization to other situations difficult (Birkmann, 2013b). Since vulnerability and risk are complex, multi-faceted and can only be measured indirectly, one needs to identify proxy data that are able to represent the component at hand.

Proxies, or indicators, reveal connections that constitute vulnerability in the absence of directly quantifiable relationships or measurements (Tonmoy et al., 2014). Indicators have the advantage of reducing the complexity of vulnerability and the aggregation of multiple indicators makes the construction of a combined vulnerability index possible (de Sherbinin et al., 2019). However, critique of the use of such composite indicators includes that they can be misleading, they are not objective and they may need large amounts of data (Feldmeyer et al., 2020). This critique is countered by a selection of studies providing

guidance for the indicator selection process (Tate, 2012; Birkmann, 2013a; Fritzsche et al., 2014; Tapia et al., 2017; Fekete, 2019). Spielman et al. (2020) define criteria by which the quality of indicators can be judged: a) theoretical consistency b) internal consistency and robustness c) practicability d) transparency e) interpretability f) relevance g) external consistence. Despite these guidelines, there is still ongoing debate surrounding the usage of aggregate indicators as well as their different stages in the construction process (Greco et al., 2019). While it is common to aggregate indicators into an index, there are

different approaches for the weights of each indicator (de Sherbinin et al., 2019). Greco et al. (2019) list data-driven and participatory methods as possible weighting schemes as well as not applying weights to the indicators, e.g. equal weights. All approaches have their strengths and weaknesses and their usage depends on the purpose of the final index. According to Birkmann (2013a) the most important functions of vulnerability and risk indicators are „awareness raising", „setting priorities" and „promoting background for action". Besides that, the process of the indicator selection and the construction of the index

need to be communicated clearly in order to understand strengths and weaknesses of any index.

Vulnerability and risk are highly spatial and, consequently, there are several literature reviews on the mapping of climate vulnerability (de Sherbinin et al., 2019). According to Preston et al. (2011), vulnerability and its subcomponents exhibit high degrees of spatial and temporal heterogeneity and help communicate the 'vulnerability of place'—the potential for harm to arise from climate change interacting with the local context. Local in this sense might be misleading, since the importance of

scale has been highlighted repeatedly (Adger et al., 2005; Füssel and Klein, 2006; Fekete et al., 2010; Preston et al., 2011). Cartographic examples for climate vulnerability assessments include the global (Verisk Maplecroft, 2015; Chen et al., 2015; Eckstein et al., 2018), regional (Greiving, 2013; Reidmiller et al., 2018) and local (Pigeon et al., 2017; Jagarnath et al., 2020)





scale. On the local scale, the heterogeneity of methodological approaches is quite high due to the strong context dependency of local vulnerability. This context includes social, political, and economic conditions and drivers including localized environ-
mental degradation and climate change, which is why climate vulnerability at the local level requires consideration of broader issues relating to sustainable development (Cutter et al., 2012). Overall, indicators are often scale dependent (Greiving, 2013, Birkmann, 2013b, de Sherbinin, 2014).

Uncertainty in climate vulnerability and risk mapping is high, which can be attributed to different sources. The diverse data sources from both natural and social sciences all contain inherent uncertainties (de Sherbinin et al., 2019), as well as the
index construction process (Tate, 2013). The inherent uncertainties in future climate predictions exist due to an incomplete understanding of the climate system with its interactions and feedbacks (a), deficits in the numerical implementation of climate processes in climate models (b), uncertainties regarding future developments of the external climate forcings (greenhouse gas emissions, solar irradiance or large volcanic eruptions) (c) and the internal climate variability on different time scales (d), which are largely due to natural fluctuations and feedbacks in the climate system (Brasseur et al., 2017). These uncertainties
are addressed regularly by usage of model ensembles, structural uncertainties, however, remain a difficulty (Parker, 2013). The uncertainties in regarding social processes are considerable and epistemic to vulnerability assessments. This includes unknown or unquantifiable aspects of vulnerability as well as well as the fact that although indicators are often known to hold a relationship to processes generating vulnerability, a quantitative description of this relationship is often lacking (Tonmoy et al., 2014). Additionally, data availability and accessibility remain challenges (Preston et al., 2011), especially when comparing
countries on a subnational level. In sum, uncertainties in vulnerability mapping are high but hard to quantify (de Sherbinin et al., 2019). Despite this, assessments that deal transparently with these uncertainties can help gain a deeper understanding of what constitutes vulnerabilities and risk in the given context.

## 2  Material and methods

The above explanations have shown that there is no uniform concept or terminology to describe what is meant by vulnerability
and risk towards climate change. We draw on the vulnerability concept developed by the UBA (Buth et al., 2017), which evolves previous conceptualizations by Füssel (2007) and IPCC (2007) and is compatible with the SREX/AR5 concept. We have modified the terminology to a small extent so that risk is derived from the following subcomponents:

- Climatic stressors; climate phenomena such as heatwaves, droughts, heavy rain etc. that cause stress to the analysed system. This is termed „exposure" in AR4 and „Hazards" in SREX/AR5.

- Spatial occurrence: describes the presence of the system potentially affected by climatic stressors in a study area (e.g., critical infrastructure, land use types). AR5 uses the term „exposure", while spatial occurrence was incorporated within sensitivity in AR4.

- Sensitivity; describes the extent to which a system (e.g., economic sector, population group, ecosystem) responds to a climatic stressor based on its characteristics.





– Impact; describes the observed or potential effect of the climatic stressor on the system, taking into account the corresponding sensitivity and spatial occurrence. AR5 considers this as "risk" without (additional) adaption

   – Adaptive capacity; a system's ability to adapt to climate change in the future through additional measures and to mitigate potential damage or take advantage of opportunities.

We realize that there has been considerable debate on the conceptual frameworks (Füssel, 2007; Bohle and Glade, 2008;
Weichselgartner, 2017; Schneiderbauer et al., 2017) most of which use the same or similar subcomponents. In line with the integrative concept by Buth et al. (2017), we regard the product of the above subcomponents as "risk" if the adaptive capacity is lacking. Approaches, which include adaptive capacity, are regarded as "vulnerability". This dichotomy is important, since adaptive capacity encompasses mainly intangible characteristics (Birkmann, 2013a) which is even more difficult to quantify than other vulnerability subcomponents. Consequently, many assessments exclude adaption from their quantification schemes.
We also take such a pre-adoption perspective on vulnerability (=risk), but at the same time have used other studies to identify adaptation and adaptation potential in different sectors (Münch, 2017; Daus, 2017; Scholze et al., 2018; Averbeck et al., 2019). Fig. 2 exemplifies our framework, which differentiates between the conceptual, indicator usage and data quality aspects of the approach.

## 2.1   Data and quality audit

An indicator selection process requires goals, which, in our case, is an understanding of socio-economic climate risks on the community scale of the TMU. In order to operationalize climate risk on the regional scale, possible relevant indicators were identified through review of existing assessments and adaption strategy literature (ESPON CLIMATE, 2011; adelphi / PRC / EURAC, 2015; Min, 2015; European Environment Agency, 2017). The basis for the development and selection of indicators is a set of quality criteria which we have adopted from Birkmann (2013a) and specified for the regional characteristics described
by Scholze et al. (2018); Averbeck et al. (2019). They comprise the following criteria: a) analytically and statistically sound, b) sensitivity to underlying phenomenon, c) validity/accuracy, d) reproducible, e) data availability, f) comparability and g) appropriate scope. These standard criteria were applied to all potential indicators, thus iteratively thinning them out to the final set of indicators (fig. 2). This quality audit is of particular importance due to the different origins of the data. Therefore, it was carried out at the four different administrative units from which they were retrieved: France (F), Germany (B, P) and
Switzerland (CH). For each of the above seven criteria, a points-based quality audit was carried out, ranking between full suitability (3 points), minor limitations (2 points) and major flaws/not available (1 point). Indicators were selected if they scored at least 16 of 21 (7*3) possible points in at least three spatial entities. This translates into a mean suitability of 75% or higher.

A total of 18 indicators met the quality criteria and was therefore assigned to the risk subcomponents. Scholze et al. (2020)
provide an in-depth discussion on the operationalization of the risk framework and the derivation of the indicators. The subcomponent combined climatic stressor comprises of seven indicators, which were derived from a 16 member regional climate multi-model ensemble. It was calculated within the EURO-Cordex initiative (https://euro-cordex.net/) and provided by the





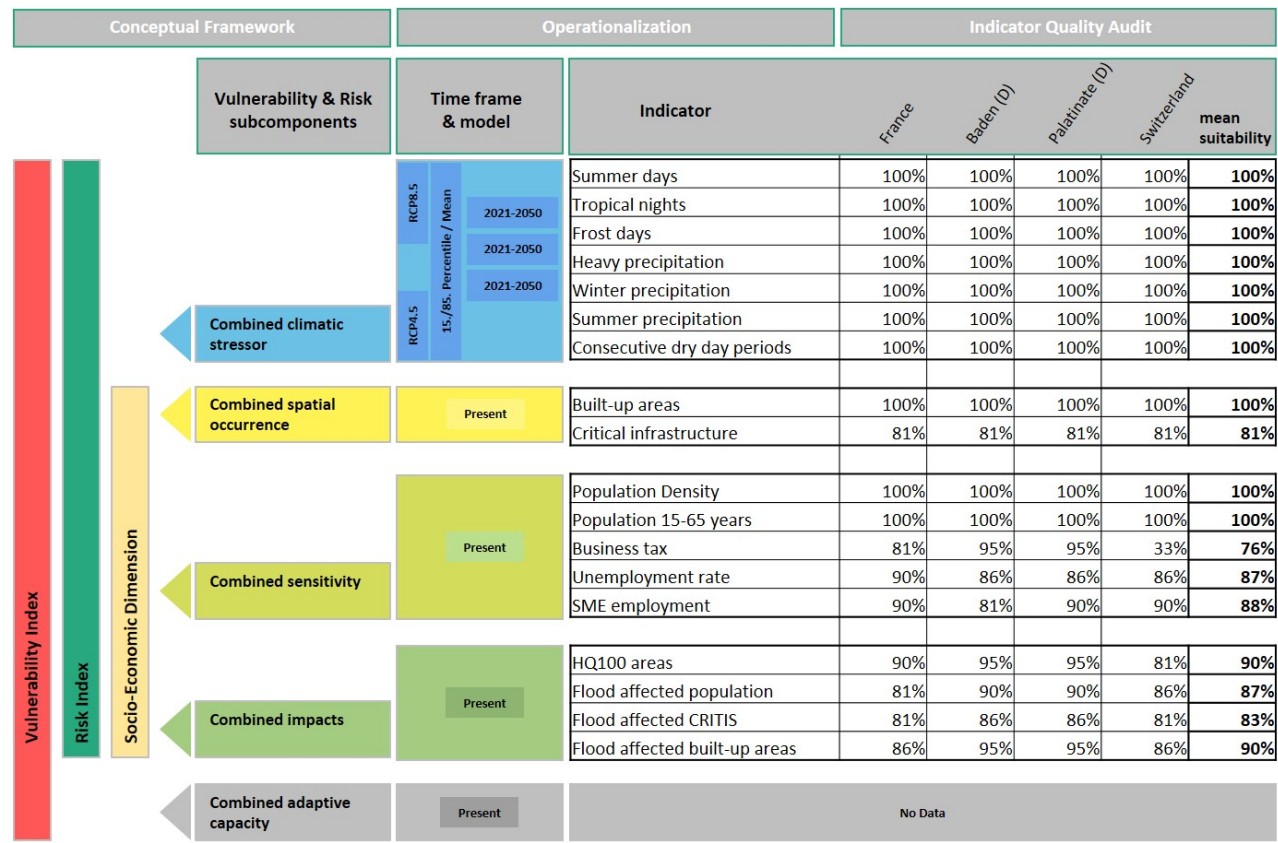

| Conceptual Framework | | Operationalization | | Indicator Quality Audit | | | | | |
|---|---|---|---|---|---|---|---|---|---|
| Vulnerability & Risk subcomponents | | Time frame & model | Indicator | France | Baden (D) | Palatinate (D) | Switzerland | mean suitability | |
| **Combined climatic stressor** | RCP8.5 / RCP4.5 — 15./85. Percentile / Mean | 2021-2050 / 2021-2050 / 2021-2050 | Summer days | 100% | 100% | 100% | 100% | **100%** | |
| | | | Tropical nights | 100% | 100% | 100% | 100% | **100%** | |
| | | | Frost days | 100% | 100% | 100% | 100% | **100%** | |
| | | | Heavy precipitation | 100% | 100% | 100% | 100% | **100%** | |
| | | | Winter precipitation | 100% | 100% | 100% | 100% | **100%** | |
| | | | Summer precipitation | 100% | 100% | 100% | 100% | **100%** | |
| | | | Consecutive dry day periods | 100% | 100% | 100% | 100% | **100%** | |
| **Combined spatial occurrence** | | Present | Built-up areas | 100% | 100% | 100% | 100% | **100%** | |
| | | | Critical infrastructure | 81% | 81% | 81% | 81% | **81%** | |
| **Combined sensitivity** | | Present | Population Density | 100% | 100% | 100% | 100% | **100%** | |
| | | | Population 15-65 years | 100% | 100% | 100% | 100% | **100%** | |
| | | | Business tax | 81% | 95% | 95% | 33% | **76%** | |
| | | | Unemployment rate | 90% | 86% | 86% | 86% | **87%** | |
| | | | SME employment | 90% | 81% | 90% | 90% | **88%** | |
| **Combined impacts** | | Present | HQ100 areas | 90% | 95% | 95% | 81% | **90%** | |
| | | | Flood affected population | 81% | 90% | 90% | 86% | **87%** | |
| | | | Flood affected CRITIS | 81% | 86% | 86% | 81% | **83%** | |
| | | | Flood affected built-up areas | 86% | 95% | 95% | 86% | **90%** | |
| **Combined adaptive capacity** | | Present | No Data | | | | | | |

**Figure 2.** Conceptual framework of vulnerability and risk, its' operationalization and quality audit for the indicators.

German Weather Service (DWD). The spatial resolution is 0.11° (EUR-11, 12.5 km), the temporal resolution includes the time horizons of the near-future (2021-2050), mid future (2041-2070) and far future (2071-2100). The three scenarios involved are

that of „moderate climate change" RCP4.5 and „business as usual" RCP8.5 (Moss et al., 2010). Taking the three time-horizons into account and analysing the 15th, mean and 85th percentile of the model ensemble in order to account for model uncertainties, each climatic stressor indicator comprises of six variations per RCP-scenario. The sum of individual stressors is therefore 126 (7 indicators * 3 time-horizons * 3 ensemble percentiles * 2 RCP-scenarios).

Data for the indicators of the spatial occurrence, sensitivity and impacts had to be researched from various German, French

and Swiss institutions and individually combined. Different national definitions of statistical parameters limited the selection for comparability reasons and language-barriers proved an obstacle. In contrast to the climatic stressors, these data refer to present day conditions. If the spatial extent was below the community level, data was summed up at the community level, for example, critical infrastructure (100*100 m resolution) was summed-up to critical infrastructures per community.

All 18 indicators, including the 126 variations of the climatic stressors, were statistically analyzed and individually mapped.

We then rasterized and normalized the layers to their respective z-scores for comparability and in preparation of the index





construction. They were then reclassified into five classes using the Jenks and Caspall (1971) algorithm. The first aggregation step was performed on the level of the risk subcomponents by equal weights summation of the classes on a pixel basis. After another reclassification into five classes, each risk subcomponent is constructed as a dimensionless index. The indices of the risk subcomponents are aggregated to the risk index by equal weights summing and reclassified into five easy-to-understand categories ranging from "low" to "high".

Data uncertainties due to missing data or data being only available for a subcomponent scale occurred for layers of the sensitivity and the impacts. The former was the case for business tax in the whole of Switzerland as well as flood data for the Swiss canton Jura. The latter was the case for the unemployment rate (Switzerland and Germany) as well as the SME employment rate (selected Communities in Germany). The decision to include these layers despite the uncertainties was based on their overall importance for the final index and their ranking in the quality audit. Consequently, locations of uncertainties are given in the maps.

## 3   Results

In the following chapter, we present the results of the climate risk assessment for the Trinational Metropolitan Area Upper Rhine Region. Due to the huge amount of data, we concentrate on the results of the risk subcomponents. Additional maps and analyses can be found in the supplement.

### 3.1   Climatic stressors

Tabel 1 lists the climatic stressors, which consist of seven individual indicators in three different periods compared to the reference period. In order to account for model uncertainties, the 15th and 85th percentiles as well as the mean of the ensemble are assessed under scenarios RCP4.5 and RCP8.5. Areal minimum, mean and maximum values are included. The strength of the change in the respective climatic stressors depends on the configuration of the model ensemble.The variability within the ensembles is higher in the near future than between the RCP4.5 and RCP8.5 scenarios. Only in the far future do the scenarios differ significantly from each other, with the RCP8.5 scenario showing a significantly wider range of the ensemble. Additionally, areal statistics point to clear spatial variances. Detailed spatio-temporal analyses are discussed by Riach et al. (2019).

Summer days (tmax > 25 °C) and tropical nights (tmin > 20 °C) are projected to increase. The former between 1 and over 60 additional days, the latter between 0 and over 40 days. The projections all clearly indicate an increase in the number of days, with higher values and variability in the far future and under Scenario RCP8.5. The spatial patterns show higher increases in the low-lying parts of the Rhine plain, areas that are already prone to heat stress. The mountainous areas show fewer changes, however might be the most affected in terms of relative change. Frost days (tmin <0 °C) behave inversely. The decrease in days ranges from 6 to around 80 days, with lowest values and highest variability in the far future under Scenario RCP8.5. The spatial pattern shows the strongest decrease in the mountainous areas, where the number of frost days currently is the highest. The low lying areas that already have substantially fewer frost days, are projected to experience a smaller decline.




**Table 1.** Areal means of the climatic stressors according to their model ensemble configuration (Scenario, percentiles, time frame).

| Climatic Stressor | Areal Statistics | RCP4.5 | | | | | | | | | RCP8.5 | | | | | | | | |
|---|---|---|---|---|---|---|---|---|---|---|---|---|---|---|---|---|---|---|---|
| | | 2021-2050 | | | 2041-2070 | | | 2071-2100 | | | 2021-2050 | | | 2041-2070 | | | 2071-2100 | | |
| | | 15. pctl | mean | 85. pctl | 15. pctl | mean | 85. pctl | 15. pctl | mean | 85. pctl | 15. pctl | mean | 85. pctl | 15. pctl | mean | 85. pctl | 15. pctl | mean | 85. pctl |
| Summer Days [Days/Year] | min | 0.7 | 5.4 | 10.5 | 1.2 | 8.4 | 15.8 | 1.9 | 9.8 | 19 | 1.2 | 5.9 | 13.9 | 3 | 11.2 | 21 | 9.7 | 23.7 | 45.4 |
| | mean | **6** | **11.1** | **15.3** | **9** | **16.6** | **23.9** | **9.3** | **17.9** | **26.4** | **5.6** | **11.2** | **17.1** | **10.8** | **20.2** | **29.9** | **22** | **37** | **54.8** |
| | max | 9.6 | 13.9 | 18.3 | 14.7 | 20.3 | 28.5 | 14.1 | 22.4 | 31.8 | 8.7 | 14.1 | 21.7 | 16.1 | 25 | 35.8 | 31.8 | 43.3 | 61.7 |
| Tropical Nights [Days/Year] | min | -0.1 | 0.5 | 0.4 | 0 | 0.9 | 0.5 | 0 | 1.4 | 1.1 | 0 | 0.6 | 0.4 | 0 | 1.4 | 1.1 | 0 | 4.7 | 5.7 |
| | mean | **0.3** | **2.1** | **5** | **0.4** | **3.6** | **8.2** | **0.6** | **4.7** | **8.8** | **0.2** | **2.3** | **4.3** | **0.6** | **5.2** | **9.2** | **2.5** | **14.2** | **23** |
| | max | 1.6 | 4.3 | 11.1 | 2.4 | 7.3 | 16 | 2.5 | 8.9 | 16.9 | 0.9 | 4.8 | 10.5 | 2.6 | 9.9 | 17.4 | 8.9 | 23.6 | 42.6 |
| Summer Precipitation [Change in %] | min | -26.5 | -8.5 | -1.4 | -29 | -11 | -4 | -28.2 | -12 | -1 | -22.7 | -0.9 | -0.1 | -25.9 | -6.6 | -5.1 | -67.7 | -34 | -7.8 |
| | mean | **-13.2** | **-4.8** | **4** | **-18** | **-6.5** | **3.4** | **-17.7** | **-5.3** | **7.2** | **-11** | **3.4** | **7** | **-16.2** | **-0.2** | **6.9** | **-43.9** | **-19** | **6.6** |
| | max | -6.4 | -0.8 | 10.6 | -7.2 | -1.8 | 10.2 | -8.4 | 0.3 | 13.7 | -4.2 | 8.1 | 20 | -5.4 | 5.9 | 21.3 | -25.7 | -7.8 | 18.1 |
| Winter Precipitation [Change in %] | min | -4.4 | 5.7 | 11.7 | -9 | 4.6 | 9.3 | -10 | 5.2 | 11.8 | -8.3 | 11.4 | 13.3 | -2.3 | 13.3 | 17 | -1.4 | 10.8 | 15.7 |
| | mean | **2.1** | **8.9** | **15.7** | **1** | **7.9** | **13.6** | **4** | **10.6** | **17.2** | **-1.7** | **14.6** | **17.3** | **4** | **17.1** | **22.4** | **9.3** | **16.8** | **23.9** |
| | max | 7.3 | 13.4 | 22.9 | 6.9 | 12.3 | 20.2 | 11.3 | 15.1 | 25.9 | 9.4 | 17.2 | 22.7 | 9.4 | 21.9 | 32 | 18.8 | 23.6 | 32.5 |
| Heavy Precipitation [Days/Year] | min | -1.7 | 0.1 | 0.5 | -2.8 | 0.3 | 0.8 | -3.2 | 0.8 | 1.2 | -2 | 0.5 | 0.9 | -3.8 | 0.7 | 1.2 | -6.2 | 1.5 | 2.2 |
| | mean | **-0.4** | **0.8** | **2.1** | **-0.2** | **1.1** | **2.4** | **0.3** | **1.7** | **3** | **0.1** | **1.3** | **2.6** | **0.4** | **1.7** | **3.2** | **0.7** | **2.7** | **4.7** |
| | max | 0.4 | 2.6 | 6.6 | 0.7 | 3.1 | 8.2 | 1.8 | 3.8 | 6.9 | 1.5 | 3.8 | 8.3 | 1.9 | 4.2 | 9 | 2.6 | 5.2 | 10.3 |
| Frost Days [Days/Year] | min | -27.7 | -22 | -17.2 | -37.7 | -30 | -23.7 | -50.7 | -40 | -32.1 | -36.3 | -25 | -19.7 | -50.3 | -41 | -34.2 | -81.5 | -70 | -59.8 |
| | mean | **-25** | **-19** | **-11.8** | **-33.5** | **-25** | **-17.3** | **-40** | **-34** | **-26.5** | **-28.2** | **-21** | **-14.2** | **-39** | **-34** | **-26.7** | **-61.9** | **-55** | **-47.4** |
| | max | -21.9 | -15 | -6.8 | -28 | -20 | -11.1 | -32.6 | -27 | -20.5 | -22.5 | -18 | -9.6 | -32.7 | -27 | -20.3 | -46.9 | -43 | -36.6 |
| Consecutive Dry Day Periods [Periods/Year] | min | -38 | -6.8 | 8 | -45 | -9.4 | 20 | -54 | -9.7 | 18 | -50 | -15 | 5 | -36 | -45 | 16 | -50 | 3.3 | 40 |
| | mean | **-13.7** | **7** | **27.1** | **-14.3** | **9.3** | **36.6** | **-18.9** | **7.5** | **31.5** | **-19.5** | **-1.2** | **19.1** | **-12.5** | **-14** | **37.5** | **-15.4** | **28.1** | **63.3** |
| | max | 3 | 15.9 | 44 | 5 | 21.8 | 56 | 7 | 19.4 | 50 | 0 | 9.2 | 35 | 8 | 5 | 55 | 26 | 49.9 | 89 |

Compared to temperature driven ones, variability of projected precipitation driven parameters is higher and the spatial pattern is very much model-dependent. The change in mean precipitation rate for the winter half year (djf in %) covers a range of values between -15% to over 30% with the majority above 10% increase. Many projections show an increase in the Rhine plain, especially in the leeward areas of the Vogues mountains. Also, many projections agree on stronger increases in the higher mountainous areas and some also to increases to the north-west of the TMU. The change in mean precipitation rate for the summer half year (jja in %) covers a range of values between 10% to below -60% with the majority below a 5% decrease. The variability between and within the different projections is even higher than that of the winter precipitation. Consequently, the spatial pattern is even more heterogeneous and in parts contradictory. Some model runs project precipitation increases in areas where others project decreases.

The change in mean annual heavy precipitation (rr >20 mm) covers a range of values between -5 to above 10 days with the majority between 0 and 5. The trend indicates an increase, negative values are mostly statistical outliers. The variability increases towards the far future, especially under RCP8.5. The spatial pattern shows an east to west gradient with higher values in the mountainous areas of the Black Forest, especially the area of highest elevation. The lee effect of the Vosges mountains is indicated through this pattern. The change in mean annual consecutive dry day periods (rr <1 mm for >5 days) covers a range of values between -50 to above 80 periods, but the majority lies between -20 and 20. A clear trend cannot be deduced since the variability is very high. The spatial patterns are consequently equally unclear and sometimes contradictory. Nevertheless, drought risk is expected to increase due to higher evapotranspiration as a consequence of higher temperatures.

The combined climatic stressors aggregate the seven individual stressors with respect to RCP scenario, time frame and ensemble percentiles. Fig. 3 displays this for the ensemble mean, the 15th and 85th percentile supplement the analysis. Com-





parability is ensured by a uniform scale, which ranges between "Low" and "High". Important conclusions can be drawn from aggregating the complex spatio-temporal climate model output. Mainly, we find that a higher climatic stress is expected under RCP8.5 than under RCP4.5. This difference is particularly evident towards the end of the century, when the deviation between the two scenarios become more prominent. However, the variability within the individual climatic parameters is also higher
here, which is consequently shown by differences between the percentiles (see also supplement). Statements on the future climatic stress are therefore strongly dependent on which scenario and which percentile of the model ensemble is used.

Consideration of the underlying stressors should be taken into account when interpreting the spatial pattern of the combined climatic stressors, because indicators may average each other out during aggregation. This is exemplified by the contrasting
spatial pattern of the mainly temperature-driven stressors, which decrease (frost days) and increase (summer days and tropical nights) to a comparable extent along an altitudinal gradient. The mainly precipitation driven stressors display a lesser clear spatial pattern and also the agreement both within the ensemble and between the stressors depends more on the particular regional model. Accordingly, the highest combined climatic stress is found where, in addition to the temperature-driven stressors, the precipitation-driven stressors also show high values. These regions of highest change occur mostly in mountainous areas and
the Hochrhein area in the Southeast of the TMU.

For the mean runs in fig. 3, differences between the RCP scenarios are evident. While RCP4.5 covers values between "Low" and "Medium", RCP8.5 stretches of the full scale from "Low" to "High". In the RCP4.5 scenario, the change signal increases over all climatic stressors. While this change signal is comparably low for the time period 2021-250, it increases towards the end of the century. Overall, aggregate climatic stress increases throughout the area, but is highest in the southern part of the study
area across all elevations (2041-2070) and in the mountainous areas along the Swiss-German border (2071-2100). The spatial pattern is driven by heavy precipitation and winter precipitation, to a lower extent also by summer days, frost days and tropical nights, especially where highest changes spatially overlap each other. For RCP8.5, the magnitude of changes in all stressors is higher compared to RCP4.5. The spatial pattern in 2021-2050 resembles that of RCP45-2071-2100 and is mainly driven by changes in summer days and winter precipitation. In 2041-2070, mainly the temperature-driven stressors increase, so the
combined climatic stress rises especially in the lowlands, whereas changes in precipitation-driven stressors modify the overall stress. By 2071-2100, temperature-driven changes remain high but additionally changes in summer and winter precipitation as well as heavy precipitation increase, so the areas with the highest values are found in the mountainous regions to the east, west and south of the study area. In addition, spatial patterns of combined climatic stressors do not coincide with the national borders, e.g. transnational areas of similar climatic exposure exist.

**3.2 Socio-economic dimension**

Fig. 4 shows individual maps of the combined spatial occurrence (a), sensitivity (b) and impacts (c), including sources and places of uncertainty, as well as the combined index (d) of these three risk subcomponents. The spatial occurrence in this study describes the presence of the system potentially affected by climatic stressors, namely built-up areas and critical infrastructure per community. Unlike the climatic stressors, the data refer to present day conditions. Highest values for the spatial occurrence
(fig. 4a) are found in the major urban areas of the TMU, namely Basel, Mulhouse, Strasbourg and Karlsruhe. "Medium-high"




**Figure 3.** Combined climatic stressors as aggregate of seven individual climatic stressors. Only the mean run of the model ensemble is shown.

occurrence is often located in the direct vicinity of large urban area but also in communities such as Colmar, Freiburg or Landau. Together with those communities of "Medium" spatial occurrence, they connect the populous centres of the region along the Rhine and its tributaries. The last two classes are mainly located in the mountainous areas of the periphery. On the right side of the Rhine, primarily "medium-low" spatial occurrences are found, on the left side of the Rhine, in the Swiss Jura the French Vosges and the German Palatinate forest mainly "low" occurrence, i.e. lowest density of settlements and infrastructure.

Sensitivity (fig. 4b) describes the extent to which a system, here the economic sector and population group, responds to a climatic stressor based on their characteristics. The highest sensitivity Levels are found in the urban areas of Karlsruhe, Freiburg and Mulhouse, followed by most other populous municipalities connecting and around the major cities in the "Medium-
**Figure 4.** Socio-Economic dimension (d) as aggregate of combined spatial occurrence (a), combined sensitivity (b) and combined impacts (c). Uncertainties due to missing data and scale issues are marked.




high" category. Cross-border clusters are also evident here, most notably in the Strasbourg-Kehl-Offenburg and greater Basel

areas. Most municipalities in France and to a lesser extent communities in Germany and Switzerland belong to the category "Medium". Categories "Medium-low" and "low" mainly occur in the two latter, with no clear spatial pattern. Uncertainties related to scale occur in the German and Swiss parts. In both areas, this is due to the unemployment rate, which is only available on a coarser scale. In Germany, additionally, the SME employment rate was not fully available at the municipal level, but in part only at the district level (NUTS-3). Uncertainties in connection with missing data exist for Switzerland due to the

lack of systematic business tax information, so the combined sensitivity might be underestimated here. Nevertheless, the basic pattern of highly sensitive urban agglomerations and lesser sensitive rural areas can be seen. In cities, population densities, unemployment rates and business taxes are higher, in rural areas the population tends to be older.

Impact (fig. 4c) describes the observed or potential effect of the climatic stressor on the system by taking Sensitivity and spatial occurrence into account. In this study, impacts are solely related to river floods, due to data availability. "High" and

"Medium-high" categories occur alongside major Rhine tributaries with highest values in the greater Strasbourg area as well as Colmar and Rastatt, all with large low-lying flood risk areas. "Medium" levels are distributed among the peripheral communities of the Black Forest and in Switzerland. In the peripheral municipalities of the Vosges and in Palatinate "Medium-low" and "Low" levels dominate. In the Swiss Canton Jura, no flood data is available so it was excluded from the analysis.

The socio-economic dimension, which is an equally weighted aggregate of the three components, combined spatial occur-

rence, sensitivity and impacts (fig. 4d), shows that the highest values are in the area of the urban centres or in the axis between them. These hot spots reflect high values in each of the three subcomponents and their respective indicators. The hot spots are surrounded by "Medium-high" communities, in the case of the greater Strasbourg and greater Basel areas they are interconnected across national borders. The "Medium" category occupies the largest area with fewer communities in the Voges and Jura areas. Here, "Medium-low" and "Low" communities have the highest proportions. The uncertainties due to missing data

and scale from the previous layers are displayed. The overlay shows that the uncertainties are more pronounced in areas that lie in the "Medium" to "Low" range. However, some uncertainties also contribute to the overall result in the other categories. Especially in Switzerland, hotspots might not be detected due to missing information on flood risks and business tax.

## 3.3  Risk index

In a final step, the socio-economic dimension is aggregated with the 18 versions of the combined climatic stressors (fig. 5),

i.e. the static socio economic situation is coupled with a changing climate. Thus, future risk is assessed for the current socio-economic situation. For the climatic stressors, the different temporal dimensions, the two RCP scenarios, and the different percentiles of the model ensemble are used, Depending on these three characteristics, the "static" elements of spatial occurrence, sensitivity and impact are overlaid and modified in the final index. Depending on which of the ensemble characteristics e.g. climatic stressors are applied, risk varies in degree and spatial pattern (fig. 5). In general, risk increases towards the end

of the century, although this is much stronger under RCP8.5 scenario. Moreover, within the same time period and scenario, the 15th percentile is lower than the 85th percentile. While under RCP4.5 the value range "low/medium-low" dominates in





**Table 2.** Number of uncertainties due to either missing data, scale issues or both.

| Number of Uncertainties | Missing Data Issues | Scale Issues | Number of Communities | Percentage of Communities |
|---|---|---|---|---|
| 0 | 0 | 0 | 898 | 48.4 |
| 1 | 0 | 1 | 117 | 6.3 |
| 2 | 0 | 2 | 364 | 19.6 |
| 2 | 1 | 1 | 414 | 22.3 |
| 3 | 1 | 2 | 3 | 0.2 |
| 6 | 5 | 1 | 61 | 3.3 |

the 15th percentile in the near future, under RCP8.5 the value ranges "Medium" to "High" occur in the 85th percentile in the distant future.

The spatial pattern shows the highest values as a characteristic of urban agglomerations, with slightly lower values in their
direct proximity. Risk ranges between "Medium" and "High" throughout the different model configurations. In the more peripheral areas, the range is higher, from "Low" to "Medium-high" depending on whether the low-end or high-end assumption of the climatic stressor is applied. The uncertainties, which are known from the socio-economic dimension, are also shown in the final index. Typically, lower values co-occur with issues of uncertainty from the socio-economic dimension, especially in the canton of Jura, but also to a lesser extent in the Palatinate, the rest of Switzerland and Germany.

## 3.4    Uncertainties

Table 2 gives an overview of the number of uncertainties resulting from missing data and/or scale issues in the communities. A majority of over 74% of communities has 0 to 2 uncertainties but none due to missing data. Around 22% of communities, including all those in Switzerland, have two uncertainties due to one missing data and one scale issue. Only 0.2% have one additional scale issue. Around 3% of communities, all in the Swiss canton Jura, have uncertainty issues on a third of all layers.
In addition to the aforementioned missing data and scale issues in the whole of Switzerland, missing flood data account for the additional four uncertainties.

It must be noted that municipalities differ in terms of size, i.e. area. for example, municipalities in France or Switzerland are often smaller than their German counterparts, and the amount of very small municipalities is higher. Estimates of uncertainties must take this into account. Furthermore, it is important to recognize that the artificial municipal boundaries often have an
impact on the values. For example, Mulhouse in France, with a population of >110.000 and a municipal area of about 22 km$^2$, has a population density of about 4868/km$^2$, whereas Freiburg in Germany, with a population of about 230.000 but a municipal area of 150 km$^2$, has a population density of 1509/km$^2$.



## 4 Discussion

A main conclusion of this study is that high risk hot spots occur mainly in urban agglomerations, which often spread across
national borders in transnational corridors (fig. 5). These results are in line with those of SREX (Field et al., 2012) and Greiving
(2013), but highlight the importance of transnational corridors. Those corridors will be the most affected areas regardless of
which climatic scenario is chosen. According to these scenarios, the risk increases dramatically towards the end of the century.
The study shows a transferable and comprehensible methodology to combine data from different national administrative bodies
in one assessment. It is a unique and challenging approach, because of uncertainties due to lack of comparability or availability
of data. Accordingly, we show at which point uncertainties influence the result. The paper therefore provides important insights
into the practical handling of imperfect data, thus advancing the discussion on the operationalization and implementation of
theoretical vulnerability and risk frameworks.

The index allows, despite different uncertainties, for the identification of places of increased risk by bringing together
various data sources derived theoretically through literature. Still, risk is conceptualized very broadly in order to be generally
applicable for all economic processes. In the sense of Preston et al. (2011), this can help start a dialogue about vulnerabilities
and risks, their meaning and causes. However, future research should focus more on different sectors in order to promote
specific risks, vulnerabilities and adaptation options (Kahlenborn et al., 2021). Furthermore, focusing stronger on interactions
between climate and society instead of aggregated indicators (Fekete, 2019) could improve our understanding and help with
specific adaption measures also beyond economic risk and vulnerability.

All utilized data underwent a systematic selection process proposed by Birkmann (2013a), which aims at reducing norma-
tivity and biases. This approach is necessary, since availability and quality of statistical data remain limiting factors (ESPON
CLIMATE, 2011; Preston et al., 2011). From the available data sets, we carefully and iteratively selected those indicators most
important to climate risks of the economic sector. The in depth justification for these data in terms of their relevance is given
by Scholze et al. (2020). We realize that the socio-economic data we used are rather general. In addition to the availability
of data in the three different countries, the guiding objective was to address economic risk as broadly as possible and being
able to include all sorts of enterprises. Our final risk index applies a territorial approach, which should be complemented by
sector-specific assessments (Scholze et al., 2018; Averbeck et al., 2019; Bohnert et al., 2021 (submitted).

Furthermore, one has to bear in mind that the climatic stressors and to some extent the impacts, project future conditions
whereas spatial occurrence and sensitivity refer to present conditions. Scenarios for societal development are still an emerging
field of research (Birkmann and Mechler, 2015) and no commonly agreed best practice regarding the downscaling of global
Shared Socioeconomic Pathways exists (O'Neill et al., 2020).

The index helps approximate and visualize risk. Although there is ongoing critique of aggregating indicators into a compos-
ite index (Tate, 2013; Tonmoy et al., 2014; Fekete, 2019; Greco et al., 2019; Spielman et al., 2020), they are applied regularly
(Tapia et al., 2017; Madajewicz, 2020), because they can capture the complex and multifaceted interactions that constitute
risk. As aggregation method we decided for an equal weights approach, which is often applied (Greco et al., 2019). Alterna-
tive weights could have been, for example, determined through a Delphi survey (Greiving, 2013), Analytic Hierarchy Process





(Chattaraj et al., 2021), Principal Component Analysis or Gaussian Mixture Models (Kim et al., 2021). The sometimes recommended statistical validation of the selection of indicators (Birkmann et al., 2021a) was not carried out, but would be desirable in future research.

In this study, uncertainty has been addressed in different ways. For the climatic stressors, the ensemble percentiles and RCP scenario selection can quantify potential future climatic changes within specific development paths. It ranges from a low-end of the scale (RCP4.5/15th percentile) to a high-end of the scale (RCP8.5/85th percentile) assumption. Variability and thus uncertainty is higher for precipitation driven stressors than temperature driven ones. For the socio-economic dimension, where future model data do not exist, only present day information is possible. Our assessment's logic is to highlight which areas

could be affected the most under present day conditions and future climate change. Here lies a strong potential to implement adaptation measures. However, this must be done taking into account the uncertainties of this assessment. In the canton of Jura, for example, the lowest risk values coincide with missing impact data and for Switzerland, no information about business tax was included. Accordingly, a principally higher risk must be assumed in this area. Generally, the index overestimates flood hazards and neglects other important climatic stressors such as hail or storms. The analysis of the uncertainties shows that they

are less significant than suggested by the cartographic representation. Rather, the number of municipalities and their location (see figs. 4 & 5) can also be used to show where and by which indicators the subcomponents underestimates the risk of a municipality. Additionally, boundary effects due to community size need to be considered.

Uncertainties related to climate models are illustrated and communicated through the model ensemble. Overall, the disclosure of uncertainties thus contributes to a better understanding of the final index. However, future research should focus

on reducing uncertainties through further data homogenization across national contexts. Although dealing with uncertainties is important from a scientific perspective, researchers need to be aware that stakeholders prefer concrete statements in order to include scientific findings in their decision making (Hackenbruch et al., 2017). Consequently, research needs to consider communication aspects. Finally, while this case study serves as a starting point for the identification of risks, further research needs to focus on the sector-specific risks as well as on the topic of cascading risks.



**Figure 5.** Risk index as aggregate of the combined climatic stressors and the socio-economic dimension. Uncertainties from the missing data and scale issues are marked on the right. The climate model uncertainties are addressed through the model ensemble.





## 5 Conclusion


In this study, we examine the spatial patterns of risk towards climate change and climatic extremes in the "Trinational Metropolitan Region Upper Rhine" (TMU). We calculate a composite risk index by combining climatic stress with present-day socio-economic data. The study shows that climatic risk can be approximated in spite of a challenging data situation due to the trinationality of the study area. Addressing the various sources of uncertainty is key to understanding the strengths and

weaknesses of the index. For the climatic data, uncertainties are accounted for via two model ensembles with different driving RCP scenarios. For the socio-economic data uncertainties exist due to availability and comparability issues in the trinational context. To estimate these uncertainties, we present a methodology that quantifies and spatially locates them. The results show differentiated spatial patterns of risk with cross-border clusters i.e. transnational corridors. Risk is amplified depending on the driving climate scenario, and increases during the course of the century, especially in the transnational metropolitan corridors

of the TMU. This study hereby highlights the necessity of future adaption measures being able to capture cross border risks in spatial planning and hints towards the need for increased transnational planning.

*Author contributions.* Conceptualization (NR, NS, RG), data curation (NR), formal analysis (NR), funding acquisition (NS, RG), investigation (NR), methodology (NR), project administration (NS, RG), software (NR), supervision (RG), validation (NR, NS, RG), visualization (NR), Writing – original draft preparation (NR), Writing – review & editing (NS, RG).

*Competing interests.* The authors declare no conflict of interest.

*Acknowledgements.* The authors gratefully acknowledge the German Weather Service (DWD, in personam Andreas Walter) for providing high-end climate projection data of the TMU.



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
