# Peer review of "Mapping Transboundary Climate Change Risk: the case study of the Trinational Metropolitan Area Upper Rhine Area"

_Natural Hazards and Earth System Sciences, 2021_

## Author Comment (AC1)

**Reply to comments of referee#1 on nhess-2021-385**

**General comments:**

Overall, this paper provides a potentially interesting study of the tri-lateral area of the upper Rhine basin. The authors adopt an indicator-based approach, and use several datasets from three different countries to assess climatic vulnerability and risk.

I have reservations about the approach that is taken, because 1) different climatic risks are combined, 2) different elements of vulnerability and risk are mixed up and sometimes even double counted, and 3) there is insufficient description of the actual calculations and actual data used, especially for the exposure and sensitivity data.

1) We see it as a challenge to combine different climatic risk since they all affect the region and the people not independently/sometimes all at the time. We first analyzed the main climatic stressors by evaluating a large model-ensemble for the region and then chose seven relevant climatic stressors. This aspect has been repeatedly discussed with experts from the German Weather Service (DWD). We see it as an advantage to be able to reflect the multitude of climatic changes and the associated complexity. We focus on the overall socio-economic dimension of risk in the TMO, so naturally, the scope of the analysis is broader than would be for a single sector.

2) We are aware of this subject and the broad theoretical and conceptual discussions. For our operationalized approach we followed the practice-oriented framework of the UBA (2017) that builds on several frequently cited publications (e.g. Füssel 2007, Birkmann 2013) and is mainly used in similar, more quantitatively oriented approaches in the German speaking context. We identified this framework out of the very rich literature because it gave us the opportunity to emphasize the main focus of the paper, namely the difficulty of identifying suitable and comparable data sets in the transnational context. For this reason, we have treated the flood related layers separately from the rest of the climatic stressors because, unlike the climatic data from the model ensemble, they have been generated from different data sources. We are aware that our initial approach exaggerates flood risks and have discussed this in the paper.

   After the reviewers comments we agree that our initial conceptual framework needs improvements/clarification. We redesigned the conceptual framework as follows (see also figure 1):
   ▪ HQ100 areas are moved to the climatic stressors section
   ▪ The flood related combined impacts are dropped to reduce double counting
   ▪ We clarified, that risk is a product hazard and vulnerability. Vulnerability consists of exposure and sensitivity.

Naturally, these major revisions have led to adjustments throughout the paper (recalculations, results) which we will resubmit.

3) The calculation is performed on the basis of simple formula Risk = Hazard*Vulnerability (Exposure + Sensitivity). We will highlight this more prominently.

In fact, we kept the rationale behind the selection of the indicators brief, since an in-depth discussion on the operationalization of our risk framework and the derivation of the indicators has been published by Scholze et al. (2020). We explicitly explain in lines 175ff that we build on this paper to address the specific challenges of the trinational situation, which has an impact on the availability, homogeneity and resolution of comparable data sets. We had split the two articles because we felt it would exceed the page numbers of one article.

We will move the Table from the supplement to the main paper and also explain more in depth why we selected each indicator and what it stands for (Tables 1 and 2).

| Conceptual Framework | | Operationalization | | Indicator Quality Audit | | | | |
|---|---|---|---|---|---|---|---|---|
| Risk subcomponents | | Time frame & model | Indicator | France | Baden (D) | Palatinate (D) | Switzerland | mean suitability |
| Hazard | Combined climatic stressor | RCP8.5 / RCP4.5 15./85. Percentile / Mean 2021-2050 / 100-year return period | Summer days | 100% | 100% | 100% | 100% | **100%** |
| | | | Tropical nights | 100% | 100% | 100% | 100% | **100%** |
| | | | Frost days | 100% | 100% | 100% | 100% | **100%** |
| | | | Heavy precipitation | 100% | 100% | 100% | 100% | **100%** |
| | | | Winter precipitation | 100% | 100% | 100% | 100% | **100%** |
| | | | Summer precipitation | 100% | 100% | 100% | 100% | **100%** |
| | | | Consecutive dry day periods | 100% | 100% | 100% | 100% | **100%** |
| | | | HQ100 areas | 90% | 95% | 95% | 81% | **90%** |
| Vulnerability | Combined Exposure | Present | Built-up areas | 100% | 100% | 100% | 100% | **100%** |
| | | | Critical infrastructure | 81% | 81% | 81% | 81% | **81%** |
| | Combined sensitivity | Present | Population Density | 100% | 100% | 100% | 100% | **100%** |
| | | | Population 15-65 years | 100% | 100% | 100% | 100% | **100%** |
| | | | Business tax | 81% | 95% | 95% | 33% | **76%** |
| | | | Unemployment rate | 90% | 86% | 86% | 86% | **87%** |
| | | | SME employment | 90% | 81% | 90% | 90% | **88%** |

*Figure 1: Revision of conceptual approach*

The authors should have provided the precise formula for constructing the indicators. In indicator construction, the normalisation between highly heterogenous and datasets that have different statistical distributions and absolute values is essential, in order to develop meaningful indicators. No word is spent on this (except "statistical analysis" in Line 179).

We put this information in the supplement because we did not want to overload the paper but agree that this information is very important. For the indicator construction we referenced Scholze et. Al. (2020). We will feature the indicator construction process more prominently in the main part of the paper and also explain more of the data treatment.

Also, I do not understand why different climatic risks are combined. Is for instance extreme wind not relevant? Why is business tax important as exposure metric for tropical nights? Why is only HQ100 used, and not also HQ50 or HQ200? Why is agricultural exposure not included, when you look at rainfall? All these choices seem completely arbitrary. A hazard-specific analysis, with all its limitations, that then combines into a single indicator would have been much more useful. The results also cannot inform any policy for adaptation, except that the urban areas stand out, but that could already have been concluded from a simple map of the area …

The rationale of combining different climatic risks was to capture the complexities that climate change poses on the overall socio-economic dimension on the local scale (communities). Identifying suitable and comparable data sets in the transnational context is a key challenge and thus naturally limits the available data sets. For example, HQ100 is the only common denominator throughout the TMO. HQ10, HQ50, HQ200 and HQextreme exist for some but not all administrative units. Tropical nights indicate Heat stress and lack of nocturnal cooling, which has effects on many economic sectors and processes. Business tax is a measure for the Economic importance of a community. On the local scale (community level), these and other data sets serve as proxies for the complex cross-sector risks in the absence of independent measures for said risk. We will deepen the theoretical reasoning behind the inclusion of data sets in order to avoid the impression of arbitrariness.

We disagree, however, that the results could have been drawn by a simple map of the region. We included a broad number of different indicators, not only population density. We discuss that we are not surprised by the result that urban areas are in general more vulnerable. However, the risk pattern presented in our final maps is not just a perfect correlation with population density. For example there are lesser populated areas in the Vosges Mountains that are ranked "medium" and some areas in the Black Forest are ranked "medium-high" although they have a low population density, and so on.
We would put it like that: the hypothesis of generally more vulnerable urban agglomerations was verified by our study, just like in other similar studies, and complemented by some exceptions and nuances resulting from the interaction of the indicators used.

We thank for all the efforts and remarks, even if we do not agree with some of them. We assume that it is more a question of different disciplines and perspectives. Sometimes the critique is beyond the intended approach.

Kind regards,
NR, NS & RG

**Detailed comments:**

Line 4: Here already the concept is mixed up: impact is a product of a single hazard (scenario), combined with exposure (what the authors here confusingly term "spatial occurrence", and sensitivity). So impact can never be an ingredient together with the former three components.
We thank the reviewer for pointing this out and revised it according to figure 1.

Figure 1: This area is in western Europe, not central Europe.
The classification of individual countries as Western or Central Europe differs depending on the context and in some works Germany, Switzerland and sometimes even Eastern France are assigned to Central Europe.

Line 55: This whole section can be shortened. The issues with definitions of vulnerability and risk are well-know, and not the main topic of this paper. These issues should have been described in single short paragraph, and then the authors could motivate and adopt a decision on the approach to be taken. The current discussion is too long, and only distracts.

We felt the need to deepen the theoretical discussion in order to explain why we followed the practice-oriented approach of the UBA (2017). After the mentioned revision, we will shorten this part accordingly.

Lines 74-75: This is not correct. Risk is also regarded in SREX and AR5 as outcome without adaptation (note the typo in "adaption").
Thank you for this comment, we will change it accordingly.

Line 80: Proxies and indicators are not the same. Proxies refer to data, that are used to approximate unobserved processes and have a unit and dimension, while an indicator is (most often) a dimensionless construct made up of some data.
Thank you for this comment, we will change it accordingly.

Line 96: "highly spatial"; highly spatial what?
Thank you for this comment. We will rephrase the sentence so that it becomes clear that risk is distributed differently in space and that maps can contribute to understanding this accordingly.

Line 99: "Local in this sense might be misleading" this is unclear.
Thank you for this comment. Maybe we did not describe our understanding of local-scale clearly enough. We mention this to address different understandings of scales. Local scale is sometimes used to describe the household scale, which we do not address. We will rephrase the sentence so it becomes more clearly that the term local is not used consistently.

Line 104: Here is seems that vulnerability is regarded as inherent property, but it is a construct. So it should be said here that all vulnerability assessments are context dependent.
Thank you for this comment, we will change it accordingly.

Lines 130-132: It is unclear to me why the authors use the term "spatial occurrence", when they mean exposure. Also sensitivity and climatic hazard have spatial occurrence and spatial properties. This adds to confusion.
The guideline for the UBA's (2017) practice-oriented approach (only available in German; https://www.umweltbundesamt.de/sites/default/files/medien/377/publikationen/uba_2017_leitfaden_klimawirkungs_und_vulnerabilitatsanalysen.pdf ) states:
*"The spatial occurrence, i.e., the presence of systems potentially affected by climatic forcing in a study region, should be explicitly examined as in the IPCC 2014 concept, for example, the number of wastewater treatment plants in the flood-prone regions of a city. It changes over time due to land use changes, for example. The term exposure should be avoided because of the different meaning in IPCC 2007 and 2014."*

After the aforementioned revision of the framework, we now adopt the term exposure to avoid confusion.

Figure 2: Why is population density a sensitivity indicator? I would think this is rather an exposure/spatial occurrence indicator. Also, why is HQ100 an impact indicator and not a climatic indicator? It also overlaps with the flood affected population etc. in the same

category. So this would be double counting. Here it becomes clear that the concept seems mixed up. Finally, what is CRITIS in the figure?

Thank you for this comment. We revised the concept as mentioned above.

There are no metrics given for the different indicators. What is the unit of "business tax" for instance, or built-up area? In many studies built up areas would also be differentiated according to density, building values, and so on.

Thank you for this comment. We will feature this information in the main body of the paper instead of the supplement.

We agree that differentiating between building density, building value and more can add interesting insights. This is, however, a question of data availability. As we show in this paper, identifying suitable and comparable data sets is inevitably determined by the lowest common denominator within the trinational context.

What are the precise sources of the data? The paper is much too short on describing the non-climatic datasets, references to the (open source) data or offices where the data were provided are not given. This is not acceptable for a research paper.

Thank you for this comment. We will feature this information in the main body of the paper instead of the supplement. Additionally we add table 1 as an overview.

*Table 1: Indicandi of the data for the the risk index*

| Risk Subcomponent | Indicator | Indicandum | Description |
|---|---|---|---|
| **Combined climatic stressor** | Summer Days | Heat stress, oppressive humidity, cooling energy demand | days/year with a maximum temperature >25 °C |
| | Tropical Nights | Heat stress, lack of nocturnal cooling | nights/year with a minimum temperature >20 °C |
| | Frost Days | Decrease of snow cover | days/year with a minimum temperature <0 °C |
| | Winter Precipitation | Flood risk in winter months | change of mean precipitation in December, January and February in % |
| | Summer Precipitation | Summer drought risk, low-water, water shortages | change of mean precipitation in June, July and August in % |
| | Heavy Precipitation | Damages caused by heavy rain and subsequent flooding | days/year with a precipitation >20 mm |
| | HQ100 areas | flood prone areas in a 100-year-event | percentage of total community area |
| **Combined exposure** | Built-up areas | Location of exposed enterprises and population | built-up area in % per community |
| | Critical infrastructure | Location of exposed critical infrastructure | incl. roads, cross-border bridges, railway lines, stations, airports, hospitals, power lines, power pylons, power towers, substations, power plants and generators |
| **Combined sensitivity** | Population density | Density of the potentially affected population | per km$^2$ |
| | Population 15-65 years | Share of the population at working age | percentage of total population |
| | Business tax | Economic importance of a community | in per , without Switzerland |
| | Unemployment rate | Economic situation of a community | percentage per community |
| | SME employment | SMEs are more sensitive due to reduced financial resources | percentage of employees in enterprises <200 (F) or <250 employees (D + CH) |

Lines 174-175: This is a too short description of the source of these data.

Thank you for this comment. We will deepen the theoretical reasoning behind the inclusion of data sets in order to avoid the impression of arbitrariness and to highlight more clearly from which sources the data originated. We will move Table 2 from the supplement to the main body of the text.

*Table 2: Data sources for the indicators of the risk index*

[revised manuscript text omitted]

Line 260: Section 3.2: Socio-economic dimension is a poor term for the various indicators included here. HQ100 areas for instance is mostly a physical variable. Also critical infrastructure and built-up areas have a mostly physical character, that is maybe influenced by some (past) socio-economic processes.

Thank you for this comment. We revised this as mentioned above.

---

## Author Comment (AC2)

**Reply to comments of referee#2 on nhess-2021-385**

**General comments:**

The article addresses a metodology to draw climate change related risk maps in a transboundary hydrological basin, taking as a case study the Upper Rhyne. The methodology is interesting and the article is well written, but despite this I think that the Authors should clarify three key aspects, before that the article might be recommended for publication:

1. The definition of risk and of its components. The Authors correctly report that many approaches are available in the literature to define risk (R) and its components. What remains unclear is the approach followed by the Authors and how the terms of hazard (H), exposure (E) and vulnerability (V) are definied and combined. As the focus is on natural risks, I would suggest to adopt the classical form R = H E V and to evidence, on the basis of the literature, why and how other authors' definitions differ from this form;

Thank you for this comment, which has also been addressed by reviewer 1. After the reviewers comments we agree that our initial conceptual framework needs improvements/clarification and will therefore perform the calculations on the basis of the simple formula Risk = Hazard*Vulnerability (Exposure + Sensitivity). We will highlight this more prominently. In our response to reviewer 1 we explain in more detail the initial rationale of our risk framework. See also figure 2 for the revised conceptual approach.

2. Climate homogeneities and risk unhomogeneities. The Authors states that in transboundary areas mapping faces the problem of harmonizing different regional data. Yet differences of regional data rather being considered a problem should in my opinion regarded to a source of information. They can be a consequence of different theoretical approaches, data collection methods, purposes of the procedure, historical risk perception. Moreover unhomogenities in risk mapping might arise also from different geological contexts (e.g. different slopes might differently react to precipitations and be differently prone to landslides, or the extradoxal area of river bends is generally more hazardous with respect to the intradoxal one) or by different population distribution (maps reported in the Supplementary material from page 20 to page 28 shade some insights on this aspect and require to be discussed with more detail in a geographical perspective). I therefore recommend (1) to deeper investigate the origin of the unhomogeneities they found in the regional risk mapping, and (2) to clearer state whether their approach homogenizes these differences by working on the original data, or it goes beyond these differences by working on different, transboundary, datasets.

The reviewer raises an important concern, which, in our opinion consist of two dimensions; namely comparability and scale. The reviewer rightly mentions reasons for differences between similar data sets depending on the (sub)national context and we agree that valuable insights can be gained from studying this. However, cross-boundary comparability of risk and its subcomponents is limited if the underlying data sets are incomparable. It is in fact the purpose of our approach to achieve comparability between national entities despite the aforementioned challenges of the trinational situation, with its impact on the availability,

homogeneity and resolution of comparable data sets. We explicitly explain in lines 175ff that we build on the paper by Scholze et al. (2020), where a deeper discussion on the issues mentioned above is provided. We had split the two articles because we felt it would exceed the page numbers of one article.

We will deepen the theoretical reasoning behind the inclusion of data sets in order to avoid the impression of arbitrariness and to highlight more clearly from which sources the data originated. We will move Table 2 from the supplement to the main body of the text in order to support this. We will provide a better explanation of how comparability is established and how the index is calculated.

We agree that different spatial patterns on a lower scale can add interesting insights. This is, however, a question of data availability. As we show in this paper, identifying suitable and comparable data sets is inevitably determined by the lowest common denominator within the trinational context. We therefore decided not to analyze data below the community scale and neglected some interesting data sets, that failed our indicator quality audit (See figure 2). The quality audit gives a measure of how suitable each indicator is in each administrative unit as well as the overall study area (lines 150-163). It addresses the inhomogeneities between the different administrative entities. The causes for the inhomogeneities are manifold and depend on the respective data sets. For example, different thresholds are being used to classify small and medium-sized enterprises (200 or 250 employees) or data such as unemployment rates are provided on different scales (community or NUTS-3 level). We realize that these imperfect data sets result in uncertainties, so we point this out throughout the paper, the figures and tables. As long as inhomogeinities in community data of different administrative origins exist, it remains a challenge to conduct transboundary assessments. This is, however, less of a problem on the NUTS-levels, which explicitly target this issue in Europe.

3. The crucial problem of arbitrariness in risk mapping. Risk mapping is a quantitative description of the potential damages or losses consquent to an adverse event. It passes through quantitative assessment and often also through classification, normalization and weighting of much different elements. In many cases these elements share the only property that they can be in some way quantified – as far as, e.g., ecosystem services are mostly not quantifyable. These procedures often introduce margins of arbitrariness which has effects on te final maps. On the other hand it is often difficult to have an estimate of the goodness of the introduced arbitrary choice. This can be done in case collected data sets of previous similar events are available. In case such data are not available the comparison of different procedure can guide the assessment of the validity of the procedure. In the lack of previous data or in the absence of the comparison with different mapping procedures, it is difficult to assess the goodness of the proposed mapping technique. The area investigated by the Authors has been urbanized for long time and it is reported that previous maps are available. At least a comparison with previous maps is recommended also to support this point.

The reviewer raises an important issue of the limitations of risk mapping approaches and composite indicators in general. We are aware that our approach aims at quantifying intangible aspects of risk, which is why we rely on indicators. We see it as a challenge to combine different climatic risks since they all affect the region and the people not independently/sometimes all at the time. We see it as an advantage to be able to reflect the

multitude of climatic changes and the associated complexity. We focus on the overall socio-economic dimension of risk in the TMO, so naturally, the scope of the analysis is broader than would be for a single sector or a single risk. The following figure illustrates this complexity.

[Figure]

*Figure 1: Schematic overview of climate change related Impacts in the study area*

Unfortunately, no previous risk assessment of a similar scope exists for the study area. We therefore rely on an in-depth literature review (Scholze et al. 2020), in which we justify the selection and operationalization of indicators. Where it is possible (e.g. RCM ensemble), we quantify uncertainties. We critically reflect on sources of uncertainty, some being inherent to risk mapping/composite indicators, others as a result of the challenging data situation in the trinational context or both. Hence, we conclude that further research is needed to improve the quality of such multi-facetted risk assessments in a transboundary context. In this sense, we see our study as a starting point for the discussion on climate change related risks in the study area. We are aware of various internal and external validation (see for example Birkmann et al. 2022[1]) approaches and discussed the approach with stakeholders and experts. In spite of the absence of risk assessments of similar scope, we will adopt the recommendation of the reviewer and strengthen the discussion on other risk assessments as a form of validating our own results.

We thank for all the efforts and helpful remarks.

Kind regards,
NR, NS & RG
* * *
[1] https://www.sciencedirect.com/science/article/pii/S0048969721051408

**Other minor comments:**

l.6 "risk can be approximated" not clear what does it mean;

Thank you for this comment. By "approximating" we point out the difficulties of capturing the intangible characteristics of risk through an index. This aims at disclosing the limitations of the approach. We would prefer to keep it in the abstract as it is, and will explain further in the main text.

l.35 and followings: here it is important to detail some expectations (and uncertainties) of the considered climate change scenarios for the area;

Thank you for pointing this out. We will revise this section accordingly (see also figure 1 (above)). We will ensure the revision compliments the analysis of the climatic scenarios in the results section.

l.55 Introduce here a definition of risk and of its components;

Thank you for pointing this out. In line with our comments above, will include the revised definition of risk here in order to clarify our risk understanding and to improve readability.

l.67 "vulnerability of the funtion of exposure…" it is not clear, all these statement should be better detailed in a framework of a reference risk definition which should be introduced before;
l.145 "vulnerability = risk": see above

Thank you for this comment, which we also addressed in the above sections. After revising the risk framework following the suggestions by reviewers 1 and 2, this section can be shortened substantially. We felt the need to deepen the theoretical discussion in order to explain why we followed the practice-oriented approach of the UBA (2017). We will also point out more clearly, that figure 2 conceptualizes the risk formula mentioned above.

| Conceptual Framework | | Operationalization | | Indicator Quality Audit | | | | |
|---|---|---|---|---|---|---|---|---|
| | Risk subcomponents | Time frame & model | Indicator | France | Baden (D) | Palatinate (D) | Switzerland | mean suitability |
| Hazard | | RCP8.5 2021-2050 | Summer days | 100% | 100% | 100% | 100% | 100% |
| | | 2021-2050 | Tropical nights | 100% | 100% | 100% | 100% | 100% |
| | | 2021-2050 | Frost days | 100% | 100% | 100% | 100% | 100% |
| | | 15./85. Percentile / Mean | Heavy precipitation | 100% | 100% | 100% | 100% | 100% |
| | | | Winter precipitation | 100% | 100% | 100% | 100% | 100% |
| | | RCP4.5 | Summer precipitation | 100% | 100% | 100% | 100% | 100% |
| | Combined climatic stressor | | Consecutive dry day periods | 100% | 100% | 100% | 100% | 100% |
| | | 100-year return period | HQ100 areas | 90% | 95% | 95% | 81% | 90% |
| | Combined Exposure | Present | Built-up areas | 100% | 100% | 100% | 100% | 100% |
| | | | Critical infrastructure | 81% | 81% | 81% | 81% | 81% |
| Vulnerability | | | Population Density | 100% | 100% | 100% | 100% | 100% |
| | | | Population 15-65 years | 100% | 100% | 100% | 100% | 100% |
| | | Present | Business tax | 81% | 95% | 95% | 33% | 76% |
| | Combined sensitivity | | Unemployment rate | 90% | 86% | 86% | 86% | 87% |
| | | | SME employment | 90% | 81% | 90% | 90% | 88% |

*Figure 2: Revision of conceptual approach*

l.199 RCP4.5 and RCP8.5: introduce a small description of the scenarios

Thank you for this comment. We will include a description of the scenarios

l.205 At which time scenario are these data referred?
Thank you for this comment. In figure 2, we highlight that we utilize two sorts of time frames. The Hazard/Combined climatic Stressors refer to future data e.g. the RCP scenarios and Flood data. The Vulnerability data refers to present day (collected) data. In line with the reply to reviewer 1, we will move Table 2 from the supplement to the main body of the text and also include the respective time frames.

l.215 and around: how was the reliablility fo the scenarios assessed? I recommend firstly to make a comparison between measured data and the simulation of present time, to identify the biases and the proper downscaling (of simulations) / upscaling (of measurements) procedures and then apply the same biasing and, if necessary, downscaling, to future scenarios;
Thank you for this comment. We are not sure if we understand you correctly. The RCP scenarios project different climatic futures depending on the atmospheric greenhouse gas concentration, which can be translated into radiative forcing levels. The IPPC is clear that the scenarios are not associated with probabilities but serve to highlight the ghg-dependant corridor of plausible possibilities.
The projections we use in this study were provided by the German Weather Service (DWD). The global circulation model (GCM) members were assessed in the Coupled Model Intercomparison Project (CMIP5), the regional climate models (RCM) were assessed by the EURO-Cordex initiative. The DWD performed a bias correction. We additionally specify the ensemble percentiles in order to account for model uncertainties.
We hope we could clarify that the models have and continue to be monitored. However, an extensive evaluation of the models' performance is beyond the intended scope of this paper and we refer to the DWD.

l.222 rr > 20 mm: what does rr stand for?
rr stands for rainfall runoff. We will write it out to be more precise here.

ll.254---255 see point 2.
Thank you for this comment. We will refer more precisely to the results in the supplement. Here we have provided detailed model results for the individual climatic stressors.

ll.338---364 it seems being more a state of the art than a discussion. Many references are presented in an intriductory way: in this section they should be more detailed commented point by point in comparison with the presented approach.
Powered by
Thank you for this comment. We will revise this section in order to discuss more clearly the strengths and weaknesses of our results in relation to the literature.